# Experimental Investigation of the Unsteady Stator/Rotor Wake Characteristics Downstream of an Axial Air Turbine

**Daniel Duda** [1,*] **, Tomáš Jelínek** [2] **, Petr Milčák** [3] **, Martin Němec** [2] **, Václav Uruba** [1,4] **, Vitalii Yanovych** [1,4] **and Pavel Žitek** [1]

1   Faculty of Mechanical Engineering, University of West Bohemia in Pilsen, 306 14 Pilsen, Czech Republic; uruba@it.cas.cz (V.U.); yanovych@kke.zcu.cz (V.Y.); zitek@kke.zcu.cz (P.Ž.)
2   Czech Aerospace Research Centre, 199 05 Prague, Czech Republic; jelinek@vzlu.cz (T.J.); nemec@vzlu.cz (M.N.)
3   Doosan Škoda Power, 301 28 Pilsen, Czech Republic; petr.milcak@doosan.com
4   Czech Academy of Sciences, 180 00 Prague, Czech Republic
*   Correspondence: dudad@kke.zcu.cz; Tel.: +420-377-638-146

**Abstract:** A feasibility study of velocity field measurements using the Particle Image Velocimetry (PIV) method in an axial air turbine model is presented. The wakes past the blades of the rotor wheel were observed using the PIV technique. Data acquisition was synchronized with the shaft rotation; thus, the wakes were phase averaged for statistical analysis. The interaction of the rotor blade wakes with the stator ones was investigated by changing the stator wheel's angle. The measurement planes were located just behind the rotor blades, covering approximately 3 cm × 3 cm in axial × tangential directions. The spatial correlation function suggests that the resolution used is sufficient for the large-scale flow-patterns only, but not for the small ones. The scales of fluctuations correspond to the shear layer thickness at the mid-span plane but, close to the end-wall, they contain larger structures caused by the secondary flows. The length-scales of the fluctuations under off-design conditions display a dependence on the area of the stator and rotor wakes cross-sections, which, in turn, depend on their angle. The obtained experimental data are to be used for the validation of mathematical simulation results in the future.

**Keywords:** Particle Image Velocimetry; turbine; turbine test facility; length-scale; PIV; turbomachinery; wake

## 1. Introduction

Due to the crucial importance of steam turbines in the current "electric age", there are a large number of scientific studies concerning the flow through these admirable machines. The flow inside an axial turbine is highly dynamical, consisting of an important periodical component, which is connected with moving wakes behind the rotor blades, and a truly turbulent component. These two components play very different roles in the flow dynamics, including mixing properties and flow stability behavior. The effect of periodicity on the stability of a laminar boundary layer has been intensively studied in the past, mainly in simplified situations using both experimental and mathematical models; see, for example, [1].

The majority of the studies on axial turbomachines have been performed using pressure measurements, which are the most robust and straight forward methods [2]. Typically, simple or complex (multi-hole) pressure probes are used in combination with the moving point measurements strategy. Subsequently, the pressure and/or velocity fields are reconstructed from the multiple point measurements. This method is suitable only for the evaluation of statistical characteristics while supposing stationarity during the whole measuring procedure, and for the application of averaging or phase averaging strategies. An example of this type of measurement can be found in the pioneering work reported

in [3] and elsewhere. The development of a fast response aerodynamic probe (FRAP) [4] allows us to resolve high frequency pressure fluctuations up to tens of kilohertz [5], similar to hot-wire anemometry, which is another of the promising possibilities characterized by high frequency response up to hundreds of kilohertz, see for example, [6].

Optical methods are very seldom used in axial machines because of the problematic access. The first attempts with optical methods applications in axial turbomachines were made only to visualize the flow—see [7]. Optical density detection methods, such as shadowgraph, schlieren and interferometry, are frequently used in turbomachinery, however only in planar configurations in linear stationary cascades—see for example, [8]. These methods are not suitable for fully 3D cases, as they perform averaging in the trans-illumination direction. Relatively easy application offers another optical method, Laser Doppler Anemometry (LDA). This method requires optical access through a single window only—see, for example, [9]. However, the LDA method is of the point measurement type and needs a traversing strategy and averaging to explore a multidimensional area, which is the same as for the pressure measurement method mentioned above. In contrast, Particle Image Velocimetry (PIV) is a typical 2D method that explores the plane of measurement; however, it requires much more complex optical access consisting of at least two perpendicular windows, one for the laser and the other for the camera. There have only been a few attempts to solve this tough technical problem. For example, in [10], the stereo PIV system was applied in combination with an endoscope for the laser light sheet for the study of the blade row interaction. Lang et al. [11] used stereoscopic PIV to explore the flow between the rotor and stator wheel. They directly observed vortices in instantaneous velocity fields and their interaction with shock waves (their experiment was transonic, while the present experiment is carried out at an isentropic stage Mach number of around 0.4). Jones et al. [12] constructed a very advanced test turbine facility equipped with the $CO_2$ PLIF system with optical access through a borescope. Yun et al. [13,14] used Stereo PIV to study the flow in a rotor shroud cavity.

A comprehensive application of the above measurement techniques is shown in the studies of Göttlich and Woisetschläger. The authors focused on experimental and numerical investigations of the evolution of turbulent flow in transonic turbines. In particular, in [15–17], the patterns of vortex emissions at the trailing edge of the stator and rotor blades, as well as the interaction of their blade rows, were assessed in detail using LDA and PIV. In [18,19], the authors used these methods to clearly record the distributed velocity, flow angle and turbulence characteristics depending on the position of the stator and rotor blades. In some cases, the authors also used a Laser Vibrometer (LV) to measure the frequency of density fluctuations in the stator wake downstream of a rotor [20,21]. However, this technique is mainly used in combination with other measurement methods such as, for example, stereoscopic PIV, which allows identification of the turbulence of the flow field during the passage of the rotor blade.

The axial air turbine model used in the present investigation is described in detail in [22,23]. It was equipped with a system of windows and mirrors in the flow section. Using the single camera PIV method, a set of instantaneous 2D velocity fields was taken, covering a finite field of view (FoV) located behind the rotor wheel [24,25]. Due to the instantaneity of velocity fields, this method offers the possibility of in-depth spatial statistics of the flow pattern.

## 2. Experimental Setup

### 2.1. Test Facility

The experimental turbine facility at the Czech Aerospace Research Centre in Prague (VZLÚ) has a single stage turbine with air as a working fluid. The test rig is part of a closed-loop aerodynamic wind tunnel. It allows the pressure ratio, Reynolds number and rotational speed of the turbine to be changed independently. The pressure ratio is controlled by the rotational speed of a twelve-stage radial compressor driven by a 1.3 MW electric motor from the company ČKD. The Reynolds number can be changed by the

pressure level, which is maintained by a system of vacuum pumps. Air temperature and humidity are controlled by inter-stage heat exchangers and a condenser dryer, respectively. The turbine stage rotational speed is controlled by a hydraulic dynamometer.

### 2.2. Turbine Stage Configuration

The turbine stage geometry represents a typical 30%-reaction stage with a nominal velocity ratio of $u/c = 0.55$ (where $u$ is the circumferential mid-span rotor velocity and $c$ the isentropic stage velocity) and a nominal stage isentropic Mach number of Ma = 0.4. The stage geometrical characterization is presented in Table 1 [22].

**Table 1.** Parameters of the stator and rotor wheel. $N$ is the number of blades, $L$ denotes the blade height and $D_h$ the hub diameter. Other parameters smoothly change along the blade height and thus they are explicitly written at the hub, at mid-span and at the blade tip. $t$ means the distance between neighboring blades, $c_h$ represents the chord length and $B_{ax}$ is its projection into the axial direction; the input and output angles $\beta_{in}$ and $\beta_{out}$ are counted with respect to the tangential direction.

|   | Unit | Stator | | | Rotor IRv2 | | |
|---|---|---|---|---|---|---|---|
| $N$ | [1] | 66 | | | 82 | | |
| $L$ | [mm] | 78.5 | | | 80 | | |
| $D_h$ | [mm] | 440 | | | 440 | | |
|   |   | hub | mid | tip | hub | mid | tip |
| $t$ | [mm] | 20.9 | 24.7 | 28.4 | 16.9 | 19.9 | 23.0 |
| $c_h$ | [mm] | 32.9 | 38.8 | 44.6 | 23.0 | 24.5 | 26.0 |
| $B_{ax}$ | [mm] | 24.3 | 29.2 | 32.6 | 22.6 | 22.1 | 21.2 |
| $\beta_{in}$ | [°] | 90.0 | 90.0 | 90.0 | 26.0 | 42.0 | 65.0 |
| $\beta_{out}$ | [°] | 15.5 | 17.0 | 15.5 | 21.7 | 20.2 | 18.1 |

The nominal rotation rate (denoted as $u/c = 0.55$ throughout this paper) is achieved at a rotational speed of $2800 \, \text{min}^{-1}$. The experiments in the underload regime, described in Section 4.5, are performed at ratio $u/c = 0.7$ and at a rotational speed of $3600 \, \text{min}^{-1}$. In the overload regime, they are performed at ratio of $u/c = 0.4$ and at a rotational speed of $2000 \, \text{min}^{-1}$.

### 2.3. Particle Image Velocimetry

The experimental technique of Particle Image Velocimetry (PIV) is already a standard tool for aerodynamic research [2,26,27]. It is based on the optical observation of small *particles* that follow the flow and are illuminated by a laser sheet. When the particles are small enough, their inertial time-scale is shorter than the flow time-scale (the ratio of which is the *Stokes number*) and therefore the particle is a good *tracer*, suitable for all particle-based measurements.

A double solid state laser New Wave Solo generates pulses of duration 5 ns at a wavelength of $\lambda = 532$ nm (which corresponds to the green color). The laser beam is defocused in one direction and slightly focused in the perpendicular direction by using a compact Lens system from the company Dantec. This laser sheet enters the turbine body through a window. Inside, it is reflected from a metal-covered mirror mounted on a small traversing system, allowing a simple change of the measuring area by shifting the mirror radially.

The FlowSense Mk II CCD camera distinguishes frames separated by a delay of 4 microseconds, depending on the measuring conditions. The trigger signal is produced by the optical gate at the rotor shaft; therefore, the frame-pairs are synchronized with the rotor angle allowing it to perform *phase averaging*. The maximum repeating frequency of frame pairs is 7.4 Hz, which is the hardware limit of the camera. The camera resolution of $2048 \times 2048$ pixels corresponds to the size of a square field of view in the range 29–32 mm (for more details, see Table 2).



**Table 2.** List of measurement planes denoted by $R_{445}$–$R_{520}$, their radial location from the axis of the turbine (note that the endwall radius is 440 mm while the radius of the blade tips is 520 mm) and the corresponding size of the field of view (FoV) of the camera focused on this plane.

| Plane | Diameter [mm] | High | FoV [mm] |
|:---:|:---:|:---:|:---:|
| $R_{445}$ | 445 | $\frac{1}{16}$ | 32.6 |
| $R_{460}$ | 460 | $\frac{1}{4}$ | 32.1 |
| $R_{480}$ | 480 | $\frac{1}{2}$ | 31.0 |
| $R_{520}$ | 520 | 1 | 29.2 |

The coordinate system is as follows: the axial coordinate is denoted by $x$, the tangential by $y$, while the radial coordinate is used only to designate the radii of measurement planes. The corresponding velocities are $u$ for the axial component and $v$ for the tangential one. The radial component is not measured. In all the figures in this paper, the dominant axial velocity points from left to right; the trailing edges of the rotor blades are just behind the left edge of the figures.

## 3. Issues Related to the Data Quality

### 3.1. Troubles with Covering Glass

The most serious problem during the measurement was that of large droplets of seeding oil dirtying the covering glass between the inner and outer region of the diffuser. The hypothetical source of these large droplets is the fine fog of oil droplets, of a size of around $\sim$(1–10) $\times 10^{-6}$ m, condensing on the surface of the fast moving rotor blades, creating a continuous film. The liquid then moves, due to the centrifugal force, to the blade tips where it detaches in the form of large droplets, which splash onto the glass. The glass first contains large areas where measurements are impossible. With time, it loses its transparency completely. Therefore, this glass was removed, leaving a large square hole in the body of the diffuser (it is the upper glass represented by a light-blue rectangle in Figure 1b). This arrangement significantly changes the geometry, therefore a deeper discussion on the trustworthiness of the result was needed. A comparison of the measured velocity fields at the tip radius, with the same flow and acquisition parameters but with and without the covering glass, is shown and discussed in our conference contribution [28], which is available as *open access*. In that contribution, we observed that the flow characteristics remain unaffected by this rude change; the magnitude of the average velocity is the most affected property.

### 3.2. Check of the Statistical Quality

The amount of obtained data is a very important issue for ensuring statistically relevant results. Generally speaking, more is better, but technically it does not need to be easy. During this study, about 700 snapshots were taken in each case. This amount is not ideally large but it might be sufficient for the convergence of statistical analysis.

To check this issue, It was decided to decrease the number of frames and to calculate the standard deviation (Figure 2a,c) and flatness (Figure 2b,d) of the tangential velocity $v$ and of the in-plane vorticity $\omega$ for smaller ensembles. The smaller ensembles were generated by halving the rest of the previous halving. Therefore, a single snapshot appeared only in one of those smaller ensembles (and, of-course, inside the original one); this scheme is sketched at the bottom of Figure 2.

The standard deviation converges fairly well, but higher moments naturally need better statistical quality, therefore they are suitable for testing it. Flatness is the fourth statistical moment calculated according to Equation (4). In Figure 2b, the flatness of the tangential velocity component is plotted; its value converges to the Gaussian limit of 3. The distribution of the radial vorticity is polynomial rather than Gaussian. Thus, the vorticity flatness might converge to a different limit, the value of which is still a subject of research [29,30].

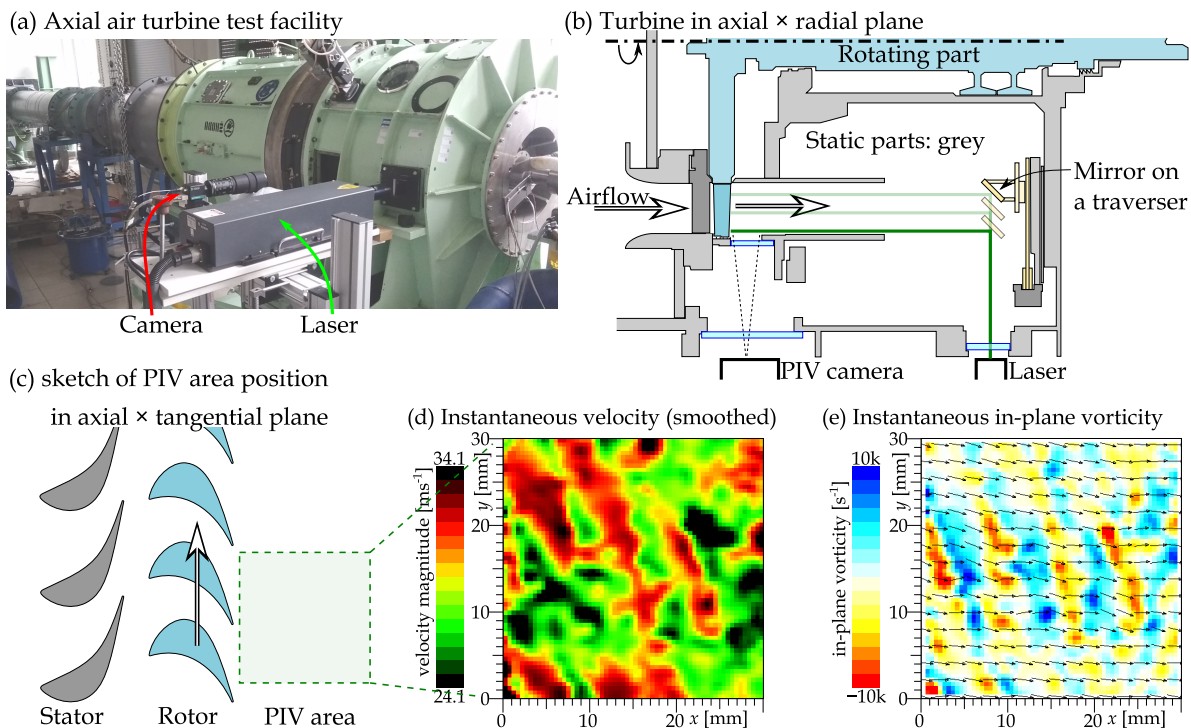

**Figure 1.** Photograph (**a**) of the test facility. The air flows from left to right. Panel (**b**) sketches the turbine in the axial × radial plane; the path of the laser sheet is manipulated by using a motorized traverser. Panel (**c**) is a sketch of the position of the area studied by PIV relative to the stator and rotor blades. Panel (**d**) represents an example of measured instantaneous velocity magnitude; for esthetic purposes, the displayed data are smoothed by a Gauss function of halfwidth $\sigma = 0.5$ mm; the legend ranges by 5 m/s around the spatio-temporal average. Panel (**e**) shows an example of the instantaneous in-plane vorticity $\omega = \Delta u / \Delta y - \Delta v / \Delta x$ under nominal conditions at $u/c = 0.55$.

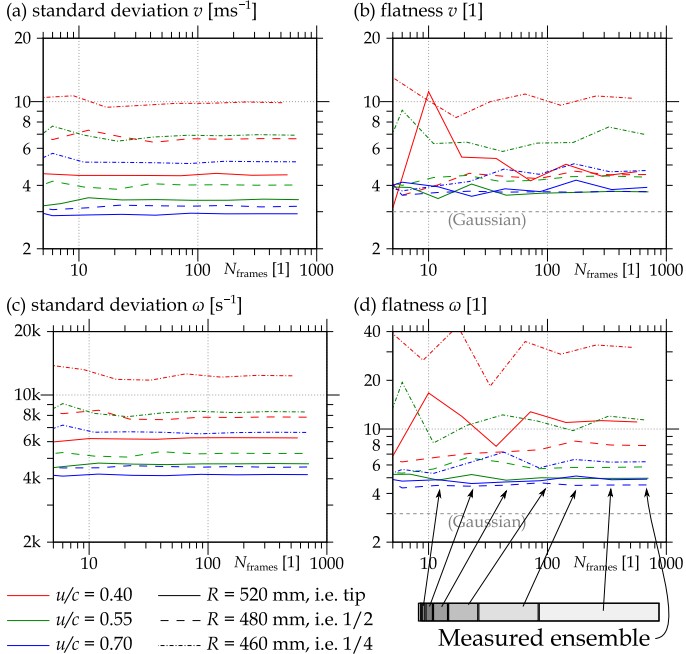

**Figure 2.** Dependence of the second and fourth statistical moment of tangential velocity and vorticity on the length $N$ of different subsets of the measured ensemble.

## 4. Results

### 4.1. Effect of Stator Wheel

The flow past a single turbine stage is affected not only by the wake of the rotor, but also by the wake of the upstream stator. The interaction of these two wake systems can be studied statistically by using the *phase averaging* of the data obtained at the same rotor angle. A slight rotation of the stator allowed us to distinguish natural waves in the rotor wakes from those caused by the stator wakes as illustrated in Figure 3, presenting the phase averaged data at three different stator wheel positions.

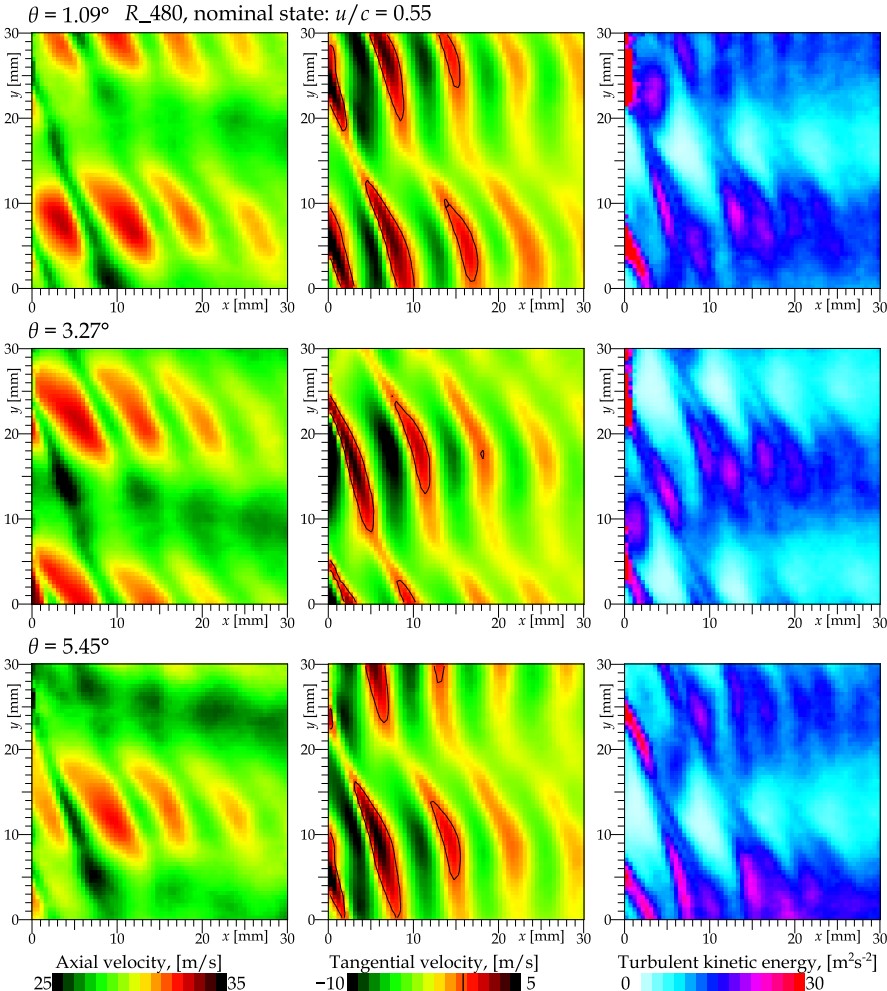

**Figure 3.** Average velocity fields obtained at different angles of the entire stator wheel to the measured area. The first row displays the data at stator angle $\vartheta = 1.09°$ to some arbitrary reference position; the second row shows data at $\vartheta = 3.27°$ and the third row at $\vartheta = 5.45°$. For the sake of clarity, the velocity vectors are not displayed, only some of the statistical quantities are displayed in color: the first column shows the ensemble average of the axial velocity component, the second shows the tangential velocity component (zero, i.e., pure axial motion, is highlighted by black isotach) and the third shows the turbulent kinetic energy. These data are measured at nominal conditions in plane $R_{480}$ at Mach number Ma = 0.4 and $u/c = 0.55$.

Analogous to the grid turbulence, we can distinguish in each wake system the following regions: the *wakes* past the blade, the free *jets* past the cascade channel and the *shear layers* in between. The velocity magnitude is lower in the wakes and it is higher in the jets. The shear regions are characterized by an increase in the turbulent kinetic energy (TKE); the shear layers quite quickly overlay the entire wake region, while the live-time of the jets is longer. Even when both are turbulized, there is a clear difference in the velocity between

the former wakes and jets. Figure 4 contains the sketch of the idea of combining systems of wakes and jets past the rotor and stator wheel as if it was a grid turbulence.

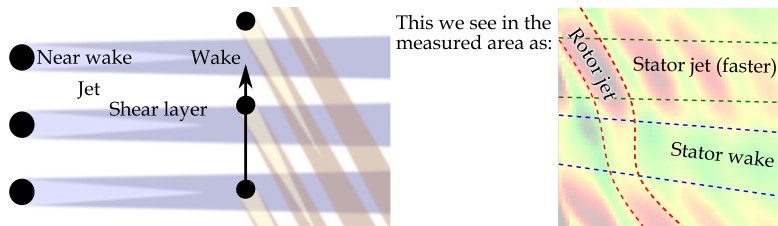

**Figure 4.** Sketch of the combination of wakes past system of static cylinders and moving cylinders. The approximate path of a jet past past a single rotor blade is depicted with red dashed lines; it is deformed as it crosses the faster and slower fluid in the stator jet and stator wake.

In Figure 3, we can easily see the systems of wake and jet structures past the rotor and stator wheel, overlaying each other. They are visible as alternating strips of higher and lower velocity. The rotor is responsible for the strips in the top left → right bottom direction, while the stator produces the wider and weaker horizontal strips.

Note that this is not just a passive addition, as can be seen from the change of the rotor strips' angle in the jets or wakes of the stator. Much more important are these effects on the evolution of vortices in both wake systems. Note that the stator wake is horizontal only at the nominal rotational speed.

The turbulent kinetic energy (TKE) shown in multiple figures in this paper is calculated by using the measured velocity components $u$ and $v$; the third (radial) component is ignored. The TKE is calculated at each spatial point by using the ensemble of local velocity components:

$$\text{TKE} = \frac{1}{2}\left(\left\langle u^2 \right\rangle - \langle u \rangle^2\right) + \frac{1}{2}\left(\left\langle v^2 \right\rangle - \langle v \rangle^2\right), \tag{1}$$

where $\langle . \rangle$ stands for ensemble averaging. Since TKE does not take into account the fluctuations in the radial direction, its value is underestimated by approximately 1/3. Additionally, fluctuations smaller than the interrogation area of the PIV method cannot be included in the TKE. On the other hand, the values of TKE are increased by the noise of the measurement, thus the estimation of something like a *true value* is not straight forward and therefore we prefer to keep the calculation procedure as simple as possible without including any artificial repair coefficients.

Under the nominal conditions ($u/c = 0.55$), the system of alternating jets and wakes past the stator can be highlighted by averaging along the axial direction, which averages out the stronger wake system past the rotor. The tangential profiles of the *axially averaged* velocity magnitude $|v| = \sqrt{u^2 + v^2}$ and the TKE are shown in Figure 5b,c. In this profile, we can select various coordinates of interest, for example, at the minimum or maximum of the velocity magnitude or at the maximum of TKE. They can be interpreted as the stator wake, stator jet and stator shear layer. The profiles of $|v|$ and TKE along the axial direction at the mentioned tangential coordinates are plotted in Figure 5d,e. These profiles follow the stator wake system, showing the wake system past the rotor: note the large difference of the velocity magnitude along the stator jet (dash-dotted blue line in Figure 5d,e). Another interesting point to mention is the double peak of TKE (Figure 5e) caused by the pair of shear layers past the rotor, which are not yet connected.

### 4.2. Radial Development

Exploring the planes at different radii in the nominal state (see Figures 6 and 7) reveals a quite homogeneous behavior with reasonably low turbulence at the tip plane $R_{520}$. At $R_{480}$, the stator wake structures become less dominant with increasing downstream distance when comparing them with the tip plane ($R_{520}$). This can be connected to the

stronger turbulence there. Closer to the hub, the secondary flow effects start to be dominant, decreasing the observability of both the stator and rotor wake structures (Figure 6) and increasing the turbulence level (Figure 7). Additionally, the quality of data at the lowest radius ($R_{445}$) is decreased due to the optical distortions caused by the focused non-flow structures (surface roughness and oil droplets) at the inner body (at higher radii, such structures are defocused due to the large distance from the focus plane).

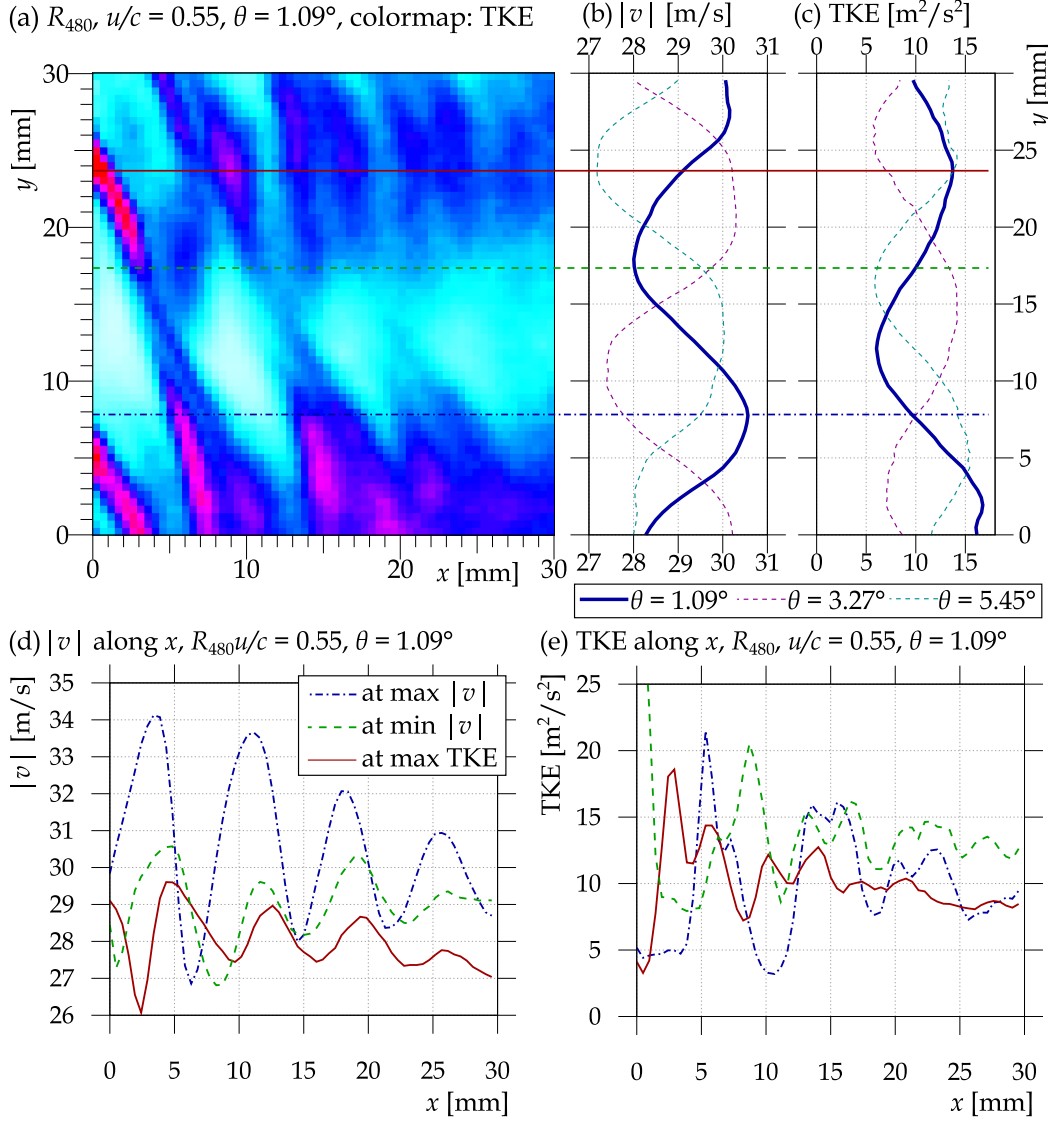

**Figure 5.** Stator wheel angle effect. Panel (**a**) shows the spatial distribution of the turbulent kinetic energy (TKE) (exactly the same data are shown in Figure 3); panel (**b**) shows the averages along the axial direction of velocity magnitude $w$; and panel (**c**) presents the TKE. For comparison, the profiles for different stator wheel angles are displayed as dashed lines. Three horizontal lines of interest are depicted by the blue dash-dotted, green dashed and solid red lines. The bottom panels (**d**,**e**) show the axial cuts of $w$ and TKE at these three lines, namely at maximum average $w$, at minimal $|v|$ and at the maximum of TKE. This data set is measured at the nominal state in plane $R_{480}$ at Mach number Ma = 0.4, $u/c = 0.55$ and stator angle $\vartheta = 1.09°$.

The anisotropy of fluctuations offers another very interesting aspect of turbulent flows. The full level of anisotropy can be investigated only in the case where we know all three velocity components [31,32], which is not our case, as we measure only the axial and tangential velocity components. Even in such a case, it is possible to evaluate the *degree of anisotropy* (DA) [33] based on two measured velocity components, judging which ones contain more fluctuations. Porreca et al. [9] present this parameter in the tangential ×

radial plane inside a multistage axial turbine. Gallego et al. [33] measure it in a wake past a single airfoil. The definition differs in the literature; therefore, we use the simple and symmetric definition of the ratio of the standard deviations of the two measured velocity components as it is used, for example, in Romano's work [34],

$$DA = \log_2 \sqrt{\frac{\langle v^2 \rangle - \langle v \rangle^2}{\langle u^2 \rangle - \langle u \rangle^2}}. \tag{2}$$

In addition to Romano [34], the logarithm function is applied, which maps the interval $(0; \infty)$ with the center in 1 to a symmetric interval $(-\infty; \infty)$ with the center in 0. Thus, in Figure 8, the value of $-1$ reads a twice larger standard deviation of the axial velocity than that of the tangential velocity and vice versa.

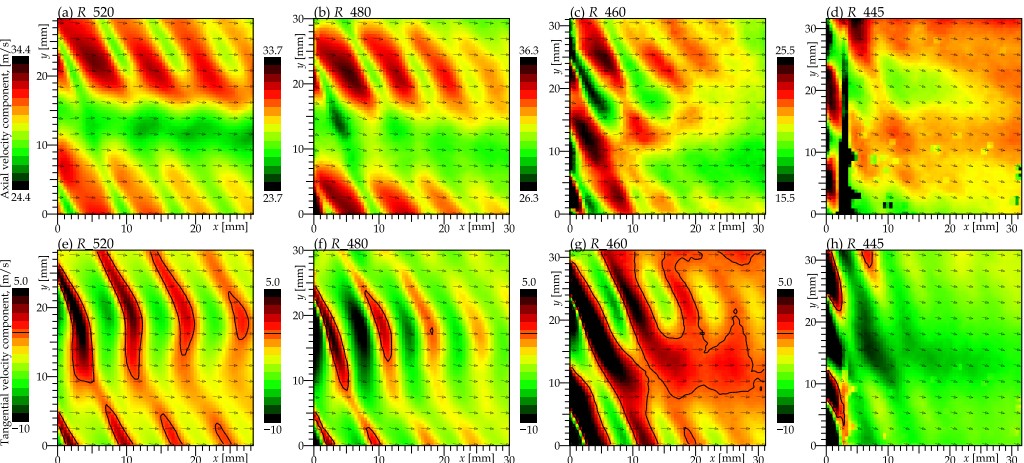

**Figure 6.** Ensemble averaged velocity at different radial planes. The colormap ranges by 5 m/s around the average axial velocity (top row), which is 29.4 m/s at the plane $R_{520}$, 28.7 at $R_{480}$, 31.3 at $R_{460}$ and only 20.5 m/s at the plane $R_{445}$. The bottom row shows the tangential velocity component with depicted zero isotach (i.e., pure axial motion). Only every fifth vector is plotted. All states are nominal, i.e., $u/c = 0.55$ at Ma = 0.4, stator angle is $\vartheta = 3.27°$.

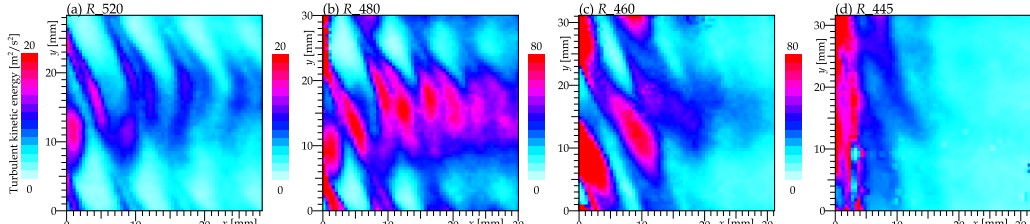

**Figure 7.** Maps of turbulent kinetic energy at different radial planes. Note that (**a**,**b**) have different color maps (0–20 $m^2 s^{-2}$) to (**c**,**d**) (0–80 $m^2 s^{-2}$). All states are nominal, i.e., $u/c = 0.55$ at Ma = 0.4, stator angle is $\vartheta = 3.27°$.

The physical interpretation, in a case where the wake is past a single obstacle, is that the dominance of traverse velocity fluctuations is typical for *bluff body* wakes, where there forms a structure of alternating large-scale von Kármán vortices [35]. This is also the case with a single airfoil or turbine blade [33]. However, this is not the case here. First of all, let us note that the DA is almost isotropic in the stator wake behind the rotor, which can be explained by the effect of isotropization during the turbulence decay [36]. Secondly, the maxima of DA are located in the area occupied by fast interblade jets. However, the maxima of DA are shifted in the axial direction with respect to the maxima of the velocity magnitude displaying the positions of jets past the rotor wheel (see Figure 9, where

a direct comparison for the mid-span plane is plotted). We are not able to explain it at this moment.

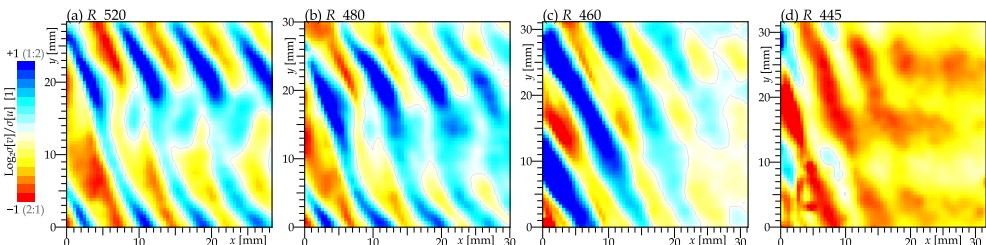

**Figure 8.** Maps of the ratio of fluctuations tangential to fluctuations in the axial direction. Bluer colors signify the dominance of tangential fluctuations over axial ones; redder colors represent areas with dominating axial fluctuations; and the white means that the fluctuations are isotropic. All states are nominal, i.e., $u/c = 0.55$ at Ma = 0.4, stator angle is $\vartheta = 3.27°$.

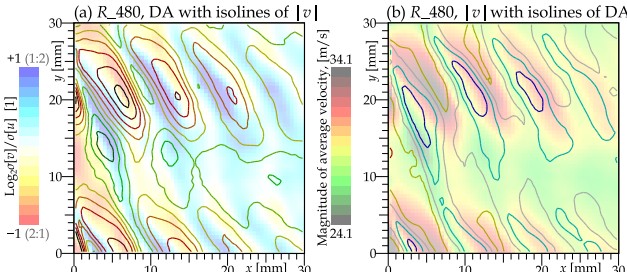

**Figure 9.** Direct comparison of spatial distribution of degree of anisotropy (DA) with average velocity magnitude $|v|$. Plane $R_{480}$, Mach number Ma = 0.4, ratio $u/c = 0.55$ and stator angle $\vartheta = 3.27°$.

Among the different planes, the spatial distribution of DA more or less follows the stripped structure observable in other quantities at higher radii ($R_{520}$ and $R_{480}$). At the plane $R_{460}$, the effect of the stator wake weakens and the effect of the rotor wake dissipates faster within this quite small FoV. Close to the hub plane ($R_{445}$), we see the dominance of axial fluctuations, which are caused by the strength of secondary flow structures, which are naturally more developed in that direction.

### 4.3. Statistics

The statistical distributions of the instantaneous tangential velocity component and instantaneous vorticity are plotted in Figure 10 for different radial planes. The data are taken from the entire field of view. In this case, which differs the most from the rest, surprisingly the plane is $R_{460}$. Its asymmetry coefficient (skewness) of tangential velocity is calculated as

$$S[v] = \left\langle \frac{(v - \langle v \rangle)^3}{\sigma^3[v]} \right\rangle, \tag{3}$$

($\sigma[v]$ is the standard deviation of $v$) and it reaches the value of 1.36 at the plane $R_{460}$, while at the other planes it is 0.48, 0.57 and 0.69 at the plane $R_{445}$. This can be caused by the small oscillations of the wake patterns, which bring more deflected velocities from the jet areas into the areas of the wakes (see Figure 11). This effect plays a role at every plane but, at higher planes, the effect of increased skewness in the wakes is almost compensated for by the opposite effect at the jets, while in the plane $R_{460}$, areas of negative skewness are smaller due to the deflection of the entire velocity field from the axial direction and thus smaller jet areas. Here some recommendation for turbine designers can arise: the wakes are too thick at this plane and the outgoing velocity could be larger in the counter-rotor direction (in order to push more momentum to the rotor). Why is the large skewness with the combination of positive average tangential velocity wrong for the turbine? Positive skewness signifies that there are few events of values larger than average, which balance a higher number of events of smaller values than average. However, the larger positive

velocities ("positive" means co-rotating with the rotor wheel) produce square times more force *against* the rotor rotation.

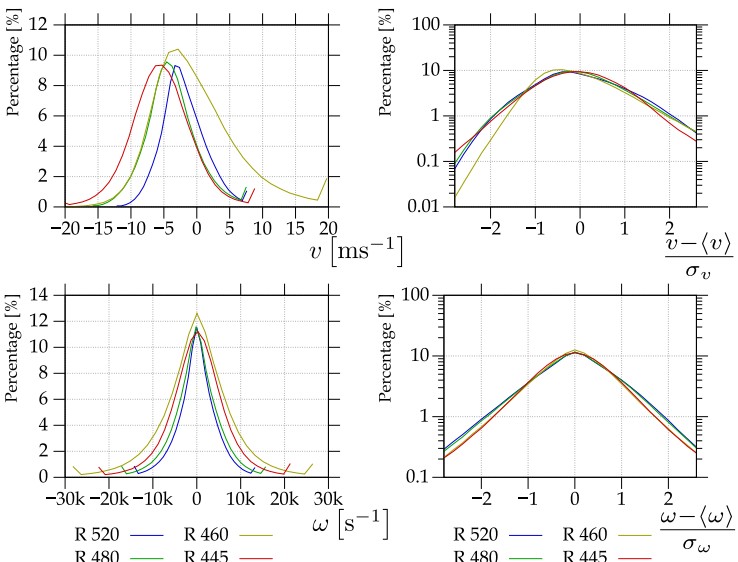

**Figure 10.** Histograms of tangential velocities (**top** row) and vorticities (**bottom** row) of the entire field of view at nominal state, i.e., $u/c = 0.55$ at Ma $= 0.4$, stator angle is $\vartheta = 3.27°$, planes are different. Right column shows the same data normalized by subtracting the average and dividing by the standard deviation in order to highlight the shape of distribution.

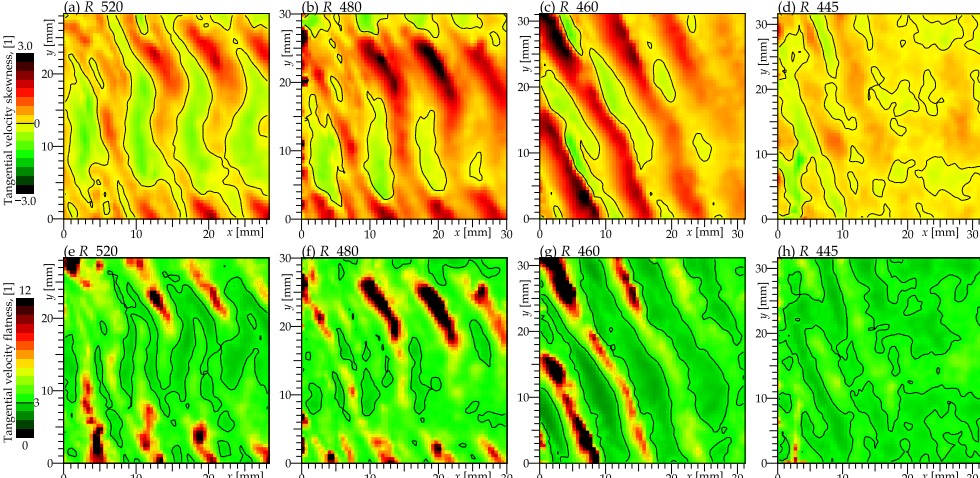

**Figure 11.** Maps of the skewness (third statistical moment) and flatness (fourth statistical moment) of the tangential velocity component. The skewness of symmetrical distribution is equal to 0, the flatness of the Gauss function is equal to 3; those values are depicted by solid black lines.

In the hub plane $R_{445}$, the flow is dominated by secondary structures, which themselves are not far from a Gaussian distribution and are distributed almost symmetrically (skewness close to zero). The departure from Gaussian distribution can be quantified by using the fourth statistical moment *Flatness* [37], sometimes called *Kurtosis*.

$$F[v] = \left\langle \frac{(v - \langle v \rangle)^4}{\sigma^4[v]} \right\rangle, \tag{4}$$

where $\sigma[v]$ is the standard deviation of $v$. $F$ is a non-dimensional quantity and it reaches the value of 3 for a Gaussian distribution and diverges for polynomial distributions. The special value of 3 is depicted in Figure 11 by a solid black line. In the case of superfluid helium,

it has been proven that a large flatness of tracer velocity is caused by the presence of highly localized quantized vortices, which change the distribution from the Gaussian to the polynomial [38]. In the case of classical turbulence, values larger than 3 are generally connected with rare strong events, which relate to *intermittent effects* [37]. At higher planes, the spots of large flatness located at the cross-sections of the rotor jet and stator jet (see Figure 12), where the flow is generally more quiet than in the wake, from time-to-time are visited by some vortex or other turbulent structure, which does not affect the TKE much due to its rareness, but significantly increases the flatness there.

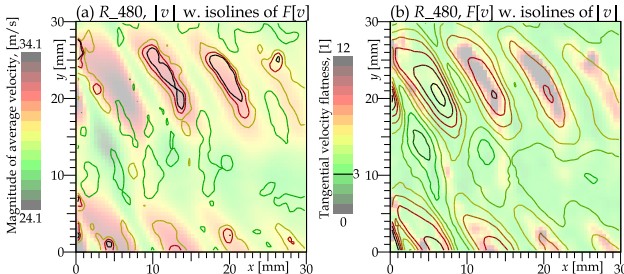

**Figure 12.** Direct comparison of spatial distribution of tangential velocity flatness ($F[v]$) with average velocity magnitude ($w$). Plane $R_{480}$, Mach number Ma = 0.4, ratio $u/c = 0.55$ and stator angle $\vartheta = 3.27°$.

The correspondence with areas of cross-sections of stator and rotor jets is supported by Figure 12, showing spots of high flatness exactly in the spots of high velocity magnitude, except for the first spot from the left, which is so close to the rotor wheel that the flow is disturbed there regularly.

Close to the hub in the plane $R_{445}$ (Figure 11d), we see more or less the Gauss value of flatness everywhere, because this area feels a high level of turbulence at all times; the strong secondary flows mix the jets and wakes completely.

We can ask not only where are the structures responsible for skewness and flatness, but also, how *big* are they? Figure 13 shows the dependence of skewness and flatness on the wave number $k$ of the fluctuations. The method for extracting fluctuations of a specific length-scale interval will be explained later. The skewness originates in larger structures (smaller $k$) for all cases, most strongly for $R_{460}$; at middle scales, the statistics are less skewed but it is still positive. At small scales (large $k$), $S[v](k)$ increases again and the values are similar, at around 0.6, except for the plane closest to the hub, whose small-scale fluctuations seem to behave similarly to those at middle scales. The flatness is increased by spatial variations; therefore, it displays values significantly larger than 3 at all scales and all planes. Smaller scales (larger $k$) produce more flatness, especially in both lower planes. At larger scales (small $k$), the flatness of tangential velocity is largest for the plane $R_{460}$. Otherwise, higher planes display less flatness of this velocity component (which also includes ensemble and spatial variations).

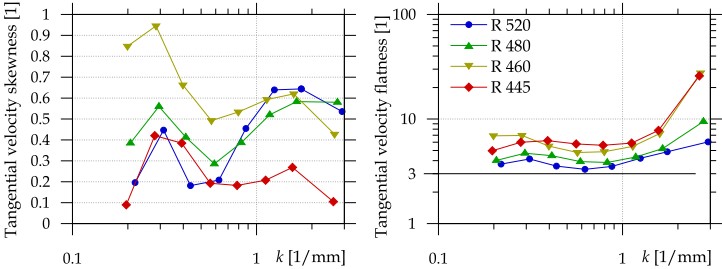

**Figure 13.** Skewness (**left**) and flatness (**right**) at different wave numbers $k$ at nominal state, i.e., $u/c = 0.55$ at Ma = 0.4, stator angle is $\vartheta = 3.27°$, planes are different. Note that the statistical moments are calculated from the entire FoV, therefore spatial variations count as well as the temporal ones.

### 4.4. Size of Fluctuation Structures

The autocorrelation function $R_{uu}(\vec{x}; \vec{y})$ [39] shows the statistical correlation of axial velocity $u$ between two positions $\vec{x}$ and $\vec{y}$ over the ensemble of $N$ points (i.e., frames in our movie):

$$R_{uu}(\vec{x}; \vec{y}) = \frac{1}{N\sigma[u(\vec{x})]\sigma[u(\vec{y})]} \sum_{t=0}^{N} u(\vec{x}, t) \cdot u(\vec{y}, t), \tag{5}$$

where $\sigma[u(\vec{x})]$ denotes the standard deviation of $u$ at point $\vec{x}$:

$$\sigma[u(\vec{x})] = \sqrt{\left\langle \left( u(\vec{x}) - \langle u(\vec{x}) \rangle \right)^2 \right\rangle}. \tag{6}$$

For the sake of convergence, the correlation function has to be run on the *Reynolds decomposed* velocity field [40–42], which means subtracting the ensemble-average velocity. The point $\vec{y}$ we chose in a fixed location at the cross-section of the stator and rotor wakes (Figure 14c) or the jets (Figure 14a), while the point $\vec{x}$ scans the entire field of view. Analogically to (5), the correlation $R_{vv}$ and the tangential velocity component are constructed and plotted in Figure 14b,d.

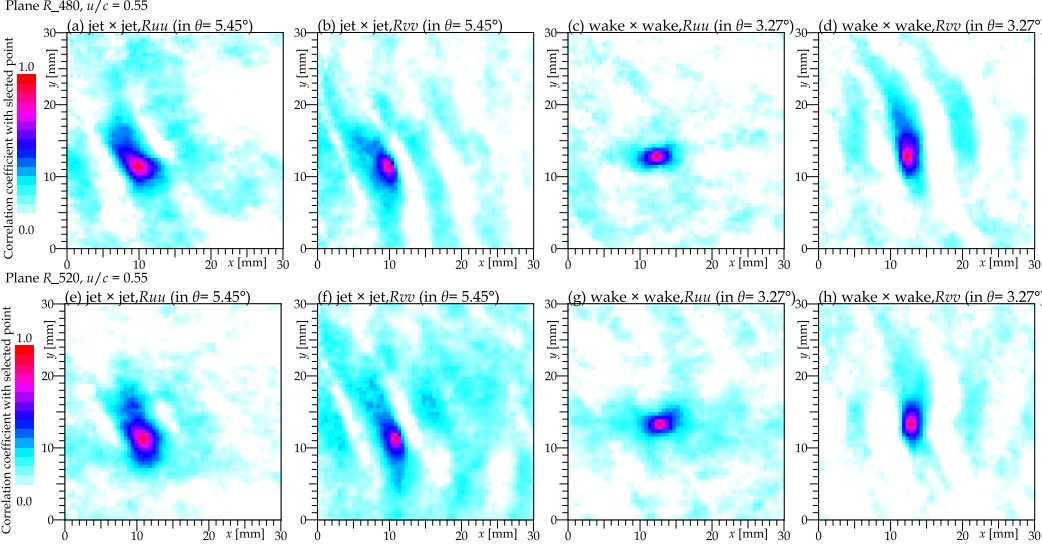

**Figure 14.** Spatial map of the autocorrelation function of the axial velocity component (panels (**a,c,e,g**)) and tangential velocity component (**b,d,f,h**) with a point at the cross-section of the stator jet and rotor jet (panels (**a,b,e,f**)), which are taken from the data sets with stator angle $\vartheta = 3.27°$ at plane $R_{480}$ (panels (**a–d**)) or $R_{520}$ (panels (**e–h**)) at $u/c = 0.55$, and at the cross-section of stator wake and rotor wake (panels (**c,d,g,h**)), which are taken from data sets with stator angle 5.45° of the same planes and regime.

Physically, it represents the area where the fluctuations are correlated with the probed point, and the size of this area refers to the typical size of turbulent structures. We see that the size of the fluctuations correspond to the cross-sectional size of the rotor blade distance at both probed locations. Additionally, a weak signal from the neighboring rotor wakes (or weaker for jets) can be observed in the spatial distribution of $R_{vv}$ (Figure 14d). This suggests the existence of structures larger than the inter-blade distance, which probably developed up-stream of the rotor wheel or even up-stream of the stator wheel.

The central peak of $R_{uu}$ in ideally resolved data would have the shape of a parabola, the half-width of which corresponds to the Kolmogorov dissipative length-scale [43–46]. In our case (see Figure 15), the peaks are sharp, suggesting that we are missing a significant part of the energy cascade towards the small-scales. This is the limitation posed by the resolution of the PIV system, which is limited by the ratio of smallest to largest resolved scales by 62, which is definitely insufficient for such a complicated flow problem. On the

one hand, we wish to see the large patterns caused by the geometry (achieved), on the other, we want to see all vortices down to the smallest ones (not achieved). A positive statement is that the current resolution is larger than the turbulent integral length-scale, therefore we can at least see the large turbulent vortices, which are responsible for the largest part of the turbulent kinetic energy.

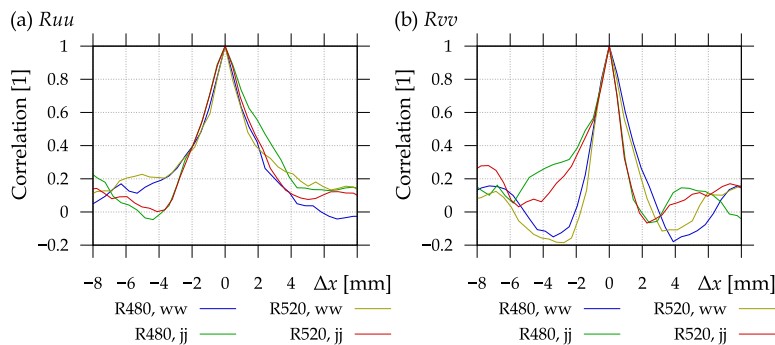

**Figure 15.** Autocorrelation $R_{uu}$ (**a**) and $R_{vv}$ (**b**) as a function of stream-wise shift from the probed point. In the legend, *ww* stands for cross-section point of stator wake and rotor wake, while *jj* abbreviates the cross-section point of the stator jet and rotor jet.

We venture to roughly estimate the integral length-scale based on *spatial* fluctuations. In hydrodynamic research, the integral length-scale $L$ is usually calculated by using time-resolved stream-wise velocities at single points. Those data are usually obtained by using the hot wire anemometry (HWA) technique and the time-series is transformed into distance by using Taylor's frozen turbulence hypothesis [47]. In our case, the correlation $R_{uu}$ between a pair of points is used as in Equation (5):

$$L[uu](\vec{x}) = \iint_{|\vec{x}-\vec{y}|<\Omega} \frac{R_{uu}(\vec{x};\vec{y})}{2\pi|\vec{x}-\vec{y}|} \mathrm{d}S, \tag{7}$$

where the $\Omega$ is the radius of the neighborhood of the probed point $\vec{x}$. The term $R_{uu}$ has to be divided by the circumference $2\pi|\vec{x}-\vec{y}|$ in order to eliminate the area growing with distance. The inconvenient property of this formula is that $L[uu]$ is not calculable everywhere in the field of view—there remains a border of undefined values, see Figure 16. Another property of this formulation is that the value at $\vec{x}$ depends on all data up to the distance $\Omega$, thus it blurs the average flow patterns.

Figure 16 shows that the variance of $L$ is not high; it fits into an interval 0.5–2.0 mm almost everywhere for the correlations of both velocity components. Note that the positions of strips of smaller and larger values do not overlay between the velocity components—which is highlighted in Figure 17—which directly compares the spatial distribution of the in-plane velocity magnitude with the estimation of the integral length-scale. The local maxima of $L[vv]$ correspond to the shear layer past the pressure side of the blade, while the distribution of $L[uu]$ roughly follows that of the velocity magnitude. The latter is caused simply by the fact that the faster velocity spreads the same fluctuating structure over a larger area.

The contributions of different length-scales to the turbulent kinetic energy are illustrated in Figure 18. That is quite a complicated and colorful plot: the different base colors (red, green and blue) represent the turbulent kinetic energy of three different size-channels: 0.5–0.8 mm for the red, 1.5–2 mm for the green and 4–6 mm for the blue channel. More details about this method, as well as about the method of spatial spectrum in Figure 19, can be found in our previous publication [48].

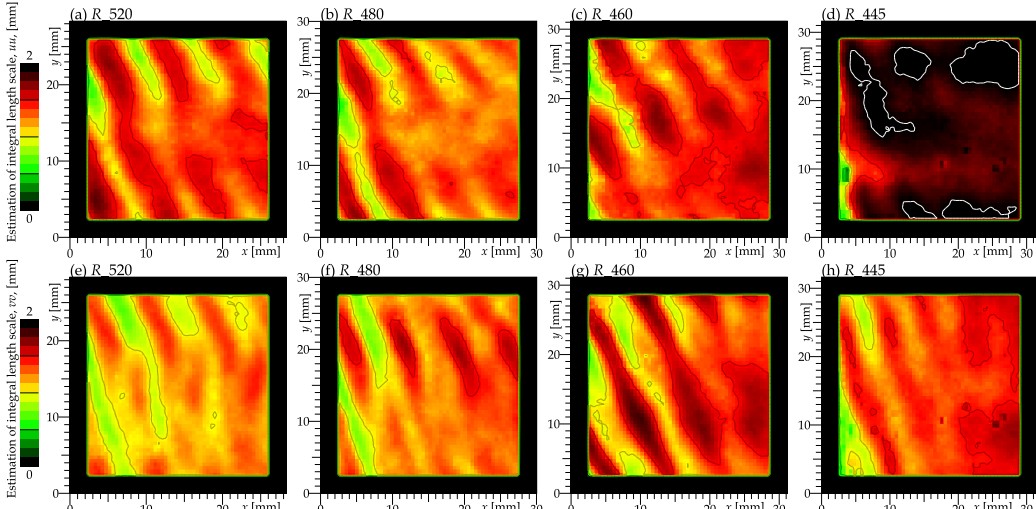

**Figure 16.** Spatial map of the rough estimation of the integral length scale of each velocity component (the top panels (**a–d**) represent the *L* of the axial component, the bottom panels (**e–h**) the tangential component). Observed values fit under 2 mm except for plane $R_{445}$, where the spots of $L[uu] > 2\,\mathrm{mm}$ are circled with white isolines. The black border around each dataset is caused by the need for a certain neighborhood along each investigated point, and such a neighborhood would not be consistent when we are too close to the edge of an FoV.

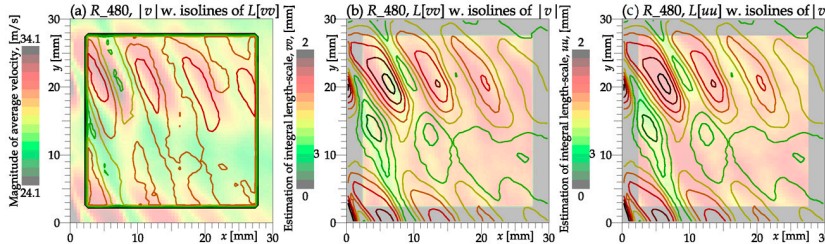

**Figure 17.** Direct comparison of the spatial map of the ensemble–average velocity magnitude $|v|$ with the estimation of integral length scales $L[uu]$ and $L[vv]$, respectively.

In panel (a) of Figure 18, we see the small-scale fluctuations mostly in the rotor wakes, which are stronger *and larger* when they meet with the stator wakes. As the flow develops, the middle scale (green) signal spreads over the faster, weakening small scale (red) fluctuations. A similar phenomenon is observable in the plane $R_{480}$, in panel (b) of Figure 18. However, at the plane $R_{460}$ (panel (c)), the secondary flows [49] lead to a pre-dominance of middle-scale structures and the rotor wakes are more apparent than in a pure TKE plot (Figure 7c). The wake displays a structure of strips of alternating *sizes* (not only intensity) of turbulence. Close to the end wall (Figure 18d), the flow is dominated by large-scale (blue) fluctuations caused by large-scale secondary flows in that area. The rotor wake structure is only weakly observable in the small-scale (red) signal. Additionally, the picture is damaged by the already discussed optical distortion causing random (thus uncorrelated and small scale) noise in the left hand side of the FoV.

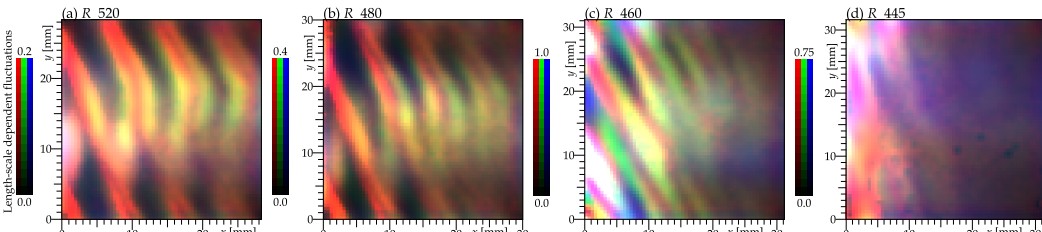

**Figure 18.** Turbulent kinetic energy colored via the length-scale of the fluctuations causing it. The smallest scales, of size 0.5–0.8 mm, are colored red; the green color represents fluctuations of size 1.5–2 mm and the blue corresponds to the largest scales 4–6 mm. Intensity units are arbitrary. Compare to Figure 7.

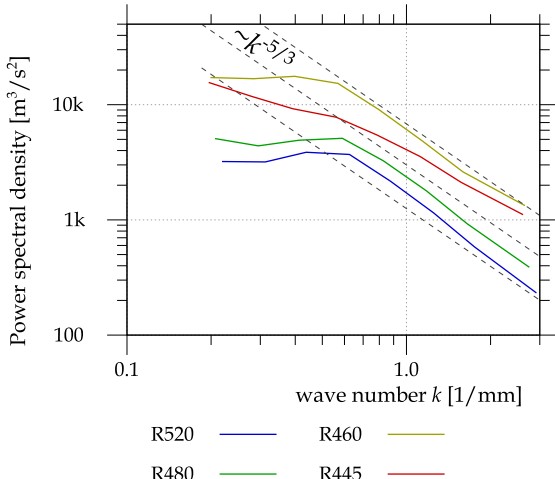

R520 ——— R460 ———

R480 ——— R445 ———

**Figure 19.** Spatial turbulence spectra for the investigated planes. The gray dashed lines represent the slope of the Richardson cascade [45].

The spatial energy spectrum (Figure 19) converges well to the Kolmogorov $k^{-5/3}$ spectrum at larger wave numbers (i.e., smaller length-scales of the fluctuations). The Kolmogorov theory is valid for ideal homogenous and isotropic turbulence [45] and is experimentally found in grid turbulence [30,50] and in any turbulence at length-scales smaller than the typical sizes of flow patterns but larger than the dissipative Kolmogorov length-scale. This fact is often used in numerical simulations to reduce the number of computational cells by substituting a so-called turbulence model into length-scales smaller than a single computational cell. Our results justify this approach, because the spectra in Figure 19 suggest that the turbulence structure follows the Kolmogorov theory, since the wave numbers are around $0.6\,\mathrm{mm}^{-1}$.

### 4.5. Off-Design Regimes

Two off-design regimes were investigated; the first, overload, at a relative velocity of $u/c = 0.4$, and the second, underload, regime at a relative velocity of $u/c = 0.7$ (note that the nominal regime corresponds to the ratio $u/c = 0.55$). The main difference lies in the large tangential component of the stage exit velocity, which turns the stator wakes in a such way that they cross the rotor wakes under a high angle ($u/c = 0.4$) or closer to the right angle ($u/c = 0.7$). The ensemble-averaged quantities at four explored radial planes past the rotor wheel are shown in Figures 20–24.

The flow conditions in regime $u/c = 0.4$ are characterized by large gradients of average tangential velocity (Figure 21) between the neighboring jets and wakes. These large gradients naturally enhance the instability and subsequent mixing of slower and faster fluid, leading to larger TKE and faster decay of these large-scale patterns. On the

other hand, in regime $u/c = 0.7$, the average velocity pattern keeps almost constant strength across the entire field of view.

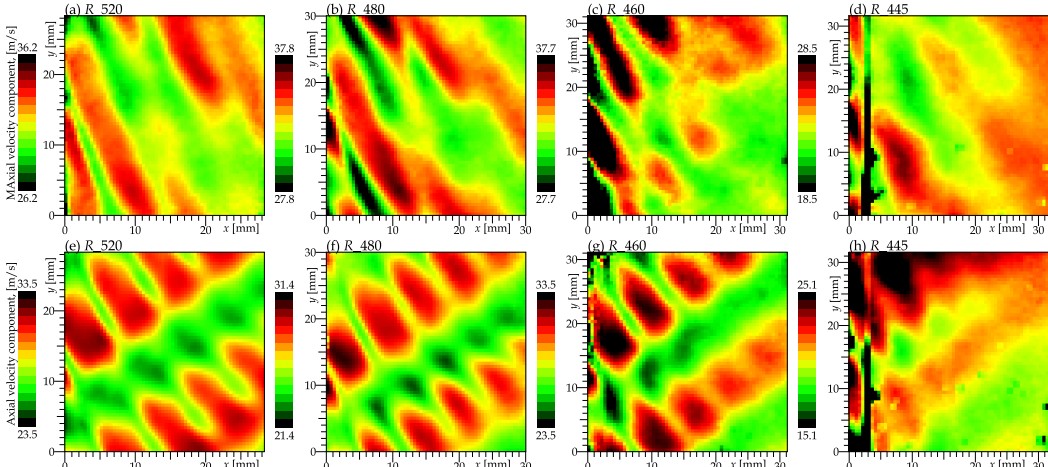

**Figure 20.** Ensemble averaged axial velocity component at off-design regimes $u/c = 0.4$ (**top** row) and $u/c = 0.7$ (**bottom** row). The color map shows the average axial velocity ranging $\pm 5\,\mathrm{m/s}$ around the FoV-averaged value.

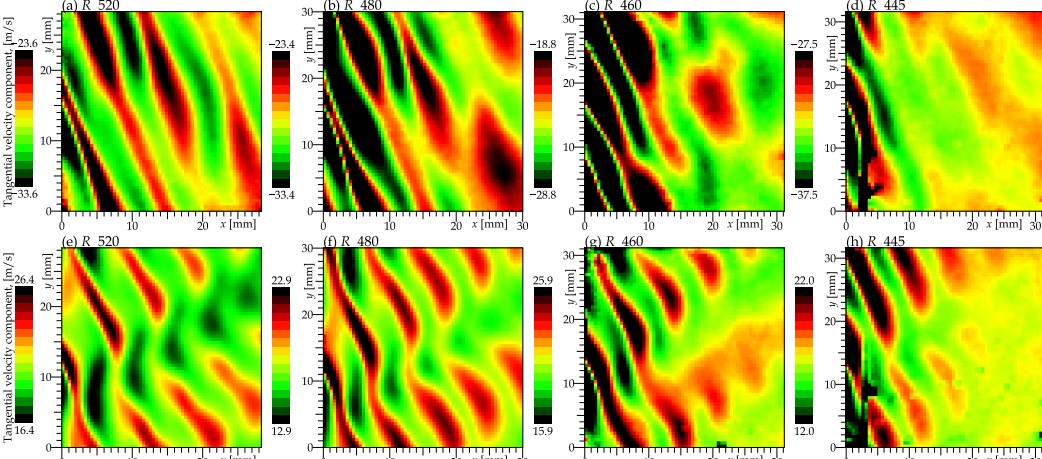

**Figure 21.** Ensemble averaged tangential velocity component at off-design regimes $u/c = 0.4$ (**top** row) and $u/c = 0.7$ (**bottom** row). The color map shows the average tangential velocity ranging $\pm 5\,\mathrm{m/s}$ around the FoV-averaged value.

The figures show that there is much more turbulence in the overload regime $u/c = 0.4$ compared to the nominal and underload regimes. This result is consistent with changes of reaction and rotor incidence angle. The overload regime had about a 15° higher incidence angle compared to the nominal regime. The stage reaction of 10% is much lower compared to the nominal regime. This combination leads to the increase of turbulence at the outlet of the rotor, presented in Figure 22, due to a weaker expansion in the rotor. A higher reaction in the case of the underload regime (the reaction was 45%) caused a decrease of the turbulence at the outlet of the rotor.

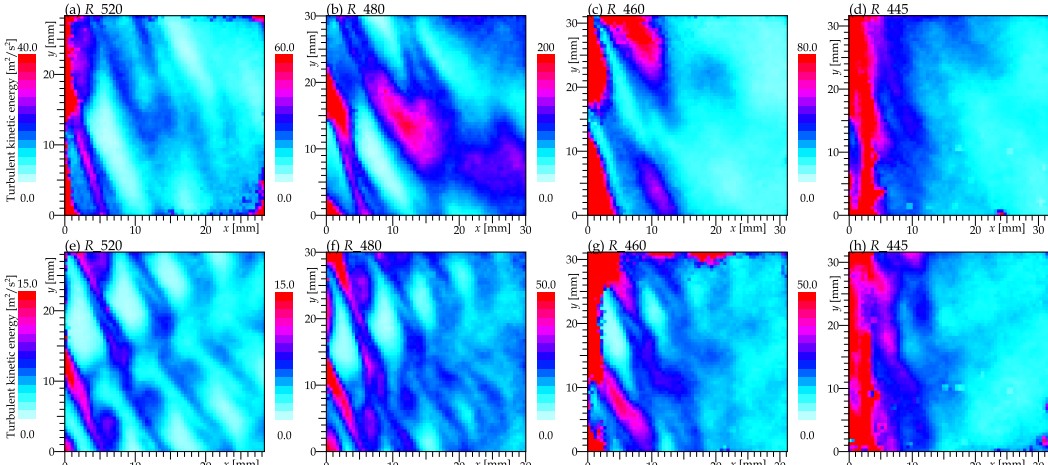

**Figure 22.** Turbulent kinetic energy based on axial and tangential velocity components (Equation (1)) at off-design regimes $u/c = 0.4$ (**top** row) and $u/c = 0.7$ (**bottom** row). The ranges of observed values vary a lot among different planes, thus the color scale is individual for each plane.

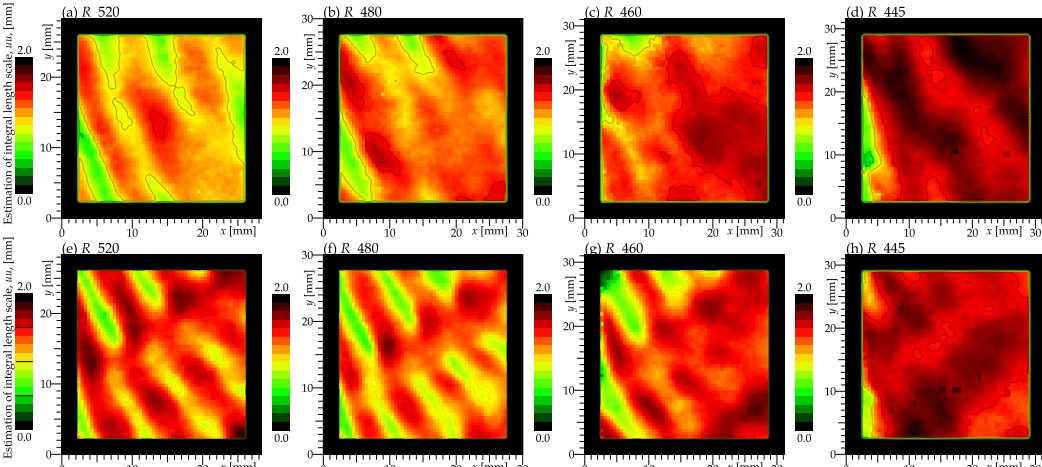

**Figure 23.** Rough estimation of the integral length scale, calculated according to Equation (7), of the axial velocity component at off-design regimes $u/c = 0.4$ (**top** row) and $u/c = 0.7$ (**bottom** row).

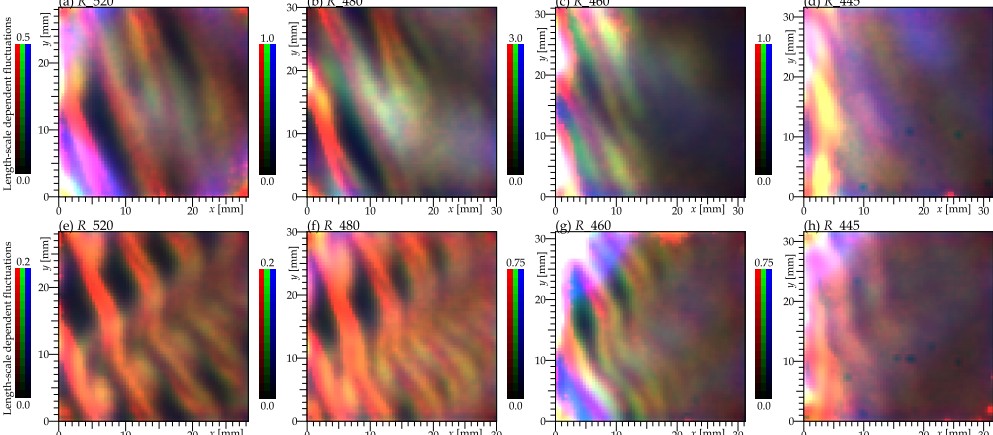

**Figure 24.** The turbulent kinetic energy colored by the length-scale of fluctuations. The scale is different among planes, but among different length-scale channels, the relative intensities are adapted to the Kolmogorov scaling. The red channel corresponds to fluctuations of the smallest size, around 0.5–0.8 mm, and the green color represents fluctuations of size 1.5–2 mm and the blue corresponds to scales of 4–6 mm.

In the first regime, the area of a cross-section of stator and rotor wakes is large due to the sharp outlet angle and therefore larger (and more energetic) structures can develop inside; it is visible in Figure 24b,c as a wide spot of middle-scale fluctuations (displayed as a green color). A buffer of large-scale-energy supports the energy at smaller scales as well via the mechanism of the Richardson cascade [51,52].

The planes closer to the hub display a larger estimated integral length-scale $L$ (Figure 23) and the turbulent kinetic energy, distinguished by the length-scale of fluctuations (Figure 24), shows stronger contributions of large-scale fluctuations than the mid-span and tip planes. Surprisingly, the estimated $L[uu]$ does not differ much between the overload and underload regimes—everywhere, we observe this quantity to lie between 1 and 2 mm with larger values towards the lower planes and increasing in the downstream direction.

The spatial spectra [48] of the off-design regimes at planes $R_{520}$, $R_{480}$ and $R_{460}$ in Figure 25 show a good scaling with the Kolmogorov law, except for large scales, where they differ in shape as well as at the point the cascade begins. An additional difference arises at small scales, where the instrumental noise plays a role, mainly at high velocities ($u/c = 0.4$). The universality is supported more by the normalization of the spectra (Figure 25 right). The normalization in this case has been done at the point in the middle of the observable part of the cascade.

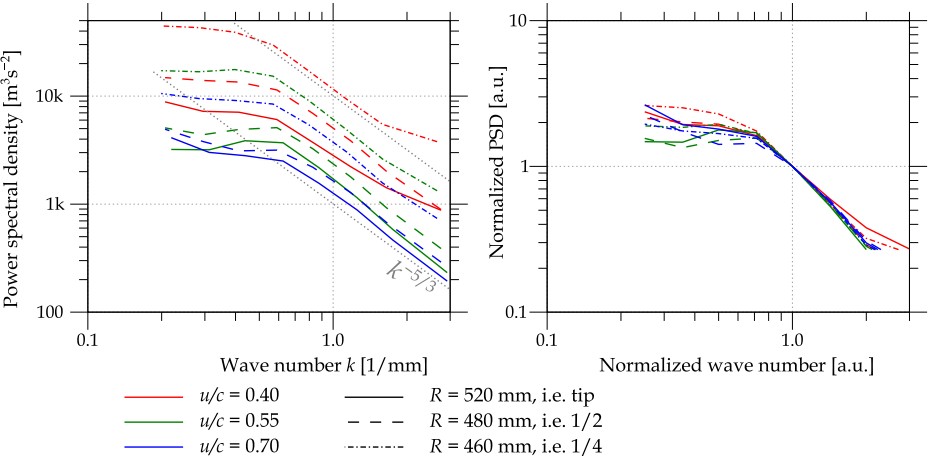

**Figure 25.** Spatial spectra of the three explored regimes depicted by colors and at three of the four explored radii distinguished via the line style. The (**right**) panel shows the same data as the (**left**) panel, normalized to the middle of the Richardson cascade.

## 5. Conclusions

This paper presents an experimental study of the wake of an axial single stage air turbine. The Particle Image Velocimetry (PIV) method has been used to observe the flow in axial × tangential planes, 3 × 3 cm, just behind the rotor blades at different radial positions. The unique design of the test turbine at the Czech Aerospace Research Centre (VZLÚ) allowed us to change the angle of the entire stator wheel in order to investigate the effect of the position of the stator wake with respect to the rotor wakes. Test were carried out under nominal conditions ($u/c = 0.55$) and two off-design regimes ($u/c = 0.4$ and $u/c = 0.7$), all at the isentropic stage Mach number 0.4.

In the investigated range of flow conditions, the following observations were made:

- The wake system of the stator blades is still observable after passing the rotor wheel, as put into evidence in the plots of the ensemble averaged velocity.
- The turbulent kinetic energy is largest at the cross-section of the shear layers of both wake systems.
- Large structures of the wakes, as well as the upper part of the inertial range, are clearly identified, but the small dissipative scales are invisible with the current setup.
- The scale of fluctuations reflects the shear layer thickness at the higher radii under nominal conditions.

- Close to the end-wall, there are flow structures of a length-scale larger than the shear layers' thickness. Therefore, the optimization of the turbine needs not only the optimization of individual blades, but also some modification of the hub geometry.
- The overload ($u/c = 0.4$) regime displays a larger scale of fluctuations caused by the larger area of cross-sections of the stator and rotor wake systems.
- The underload regime ($u/c = 0.7$) contains less turbulent kinetic energy, probably due to the smaller area of wake cross-sections, although the wakes themselves are wider as the angle of attack is not ideal for both off-design regimes. As the TKE is smaller, the wake pattern is not mixed and survives longer, which can have a negative effect on the next stage.

*Future Research*

Future investigations will focus on the two- or three-dimensionality of the flow at various radii, which can be carried out by exploring different planes (e.g., radial × tangential) or by upgrading to the stereo PIV system [53] using a pair of cameras observing the scene under slightly different angles.

The contemporary development of turbines relies on numerical simulations [23,54,55]; therefore, it would be interesting to compare our experimental results with some of them.

**Author Contributions:** Conceptualization, P.M.; methodology, V.U., D.D. and V.Y.; software, D.D.; formal analysis, T.J.; investigation, D.D.; resources, M.N.; data curation, D.D.; writing—original draft preparation, D.D.; writing—review and editing, V.U.; visualization, D.D.; supervision, V.U. and P.M.; project administration, P.Ž., P.M. and T.J.; funding acquisition, P.Ž. All authors have read and agreed to the published version of the manuscript.

**Funding:** This work originated within the framework of the project TN01000007-NCE.

**Institutional Review Board Statement:** The study was performed in conformity with the Etický kodex Západočeské univerzity v Plzni, https://www.zcu.cz/rest/cmis/document/workspace://SpacesStore/543b1541-3680-4e66-ad5e-a0db59174299;1.1/content (accessed on 10 August 2020).

**Informed Consent Statement:** Not applicable.

**Data Availability Statement:** The rough instantaneous data are available in csv. format in https://uloz.to/file/2x1Rb8197dAg/piv-measurement-of-uwb-at-vzlu-rotor-irv2-zip#!ZGtjZGR2ATH5MwVmMwEzZ2R4ZmuuMJqOMQp4MJyuGRkFMGZjMN== (accessed on 10 August 2020) (3 GB).

**Acknowledgments:** We thank Bohumil Laštovka and Jakub Pokorný for valuable technical help. We also thank Jan Uher from Doosan Škoda Power for valuable discussions preceding this measurement campaign. This work originated within the framework of the project TN01000007-NCE. The presented work was financially supported by student project SGS-2019-021 (Improving the efficiency, reliability and service life of power machines and equipment 5).

**Conflicts of Interest:** The authors declare no conflict of interest.

## List of Symbols and Abbreviations

| Symbol | Meaning |
| --- | --- |
| $\beta$ | nominal stage angle (relative to tangential direction). |
| $c$ | isentropic stage velocity. |
| DA | degree of anisotropy, $\mathrm{DA} = \log_2 \frac{\sigma[v]}{\sigma[u]}$. |
| $F[x]$ | flatness i.e., fourth statistical moment of variable $x$, $F[x] = (x - \langle x \rangle)^4 / \sigma^4[x]$. |
| FoV | field of view. |
| $k$ | wavenumber, unit $[\mathrm{mm}^{-1}]$. |
| Ma | isentropic Mach number. |
| PIV | Particle Image Velocimetry. |

| | |
|---|---|
| $R_{\mathrm{xxx}}$ | measuring plane in axial$\times$tangential direction; xxx denotes the *diameter*, if it was a cylinder. |
| $R_{uu}$, $R_{vv}$ | autocorrelation function of $u$ or $v$ velocity components respectively. |
| $L[uu], L[vv]$ | rough estimation of the integral length-scale of $u$ or $v$, $L[uu](\vec{x}) = \iint \frac{R_{uu}(\vec{x};\vec{y})}{2\pi|\vec{x}-\vec{y}|}\mathrm{d}S$ |
| $\sigma[x]$ | standard deviation of variable $x$, $\sigma[x] = \sqrt{\langle x^2 \rangle - \langle x \rangle^2}$. |
| $\vartheta$ | turn angle of the stator wheel. |
| TKE | turbulent kinetic energy. |
| $u$ | circumferential mid-span rotor velocity. |
| $u$ | axial velocity component (parallel with $x$-axis). |
| $v$ | tangential velocity component (parallel with $y$-axis). |
| $|v|$ | in-plane velocity magnitude, $|v| = \sqrt{u^2 + v^2}$. |
| $\omega$ | in-plane vorticity, $\omega = \Delta u / \Delta y - \Delta v / \Delta x$. |

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
