# Peer review of "Experimental Investigation of the Unsteady Stator/Rotor Wake Characteristics Downstream of an Axial Air Turbine"

_ijtpp, doi:10.3390/ijtpp6030022_

Round 1

Reviewer 1 Report

  • the purpose or meaning of your work has not become clear. What are you intending to do with your results?
  • There is no proper explanation or analysis of your findings nor any indication of what can be deduced.
  • The color plots are not presented appropriately, you do not give any indication of TE-locations, stator wake angles, etc.
  • It is quite awkward that the problems and restrictions of the experimental work are presented after the results. This should be done the other way round.

Author Response

Response to reviewers

First of all, we want to thank all reviewers for their work with reading our manuscript and for their comments.

Second, we want to apologize to editors and reviewers for late response.

We modified the introduction and we added an interesting observation about degree of anisotropy and the spatial distribution of flatness (fourth statistical moment) of tangential velocity.

Two of three reviewers wish the “discussion” section in front of the “results” section. We exchanged them, and we renamed it as “Issues related to the data quality”.

We added the list of symbols and abbreviations.

The application of this measurement for the process of turbine design is under development with collaboration with our industrial partner. At this moment, the only direct benefit serves for validation of their numerical simulations.

We added an artistic combination of data and manual sketch of the system of stator and rotor wakes into the footnote 2. We do not want to disturb any figure containing data, but we feel that it is important to sketch what we are talking about across the entire paper.

On behalf of author collective

Daniel Duda

Reviewer 2 Report

The paper documents an experimental research on a laboratory turbine stage, representative of a steam turbine stage. The study is interesting from the aerodynamic point of view, and the measurements performed have great scientific merit, but the paper is nearly totally focused on the measurement device and the instrument issues, with very little space left to turbomachinery aerodynamics. Authors discuss about the integration between stator and rotor wakes, but there is little attempt to provide an interpretation of the observed features in terms relevant to the typical issues of turbomachinery (for example, no idea is given on the stator-rotor interaction, stator-rotor blade number, ecc). Also, no information on the stator-exit flow field is available, so it is difficult to understand why the residual traces of the stator wakes at the exit of the rotor are so prominent.

On the other side, the paper proposes very interesting quantitative discussions on the physical character of the wakes and their turbulent properties, demonstrating the very high level of competence of the authors in such more basic fluid-dynamic aspect, which anyway are of interest also for readers working on the most basic aspects of turbomachinery flows.

So, in the end this paper can make acceptable after the authors have included more information and proposed specific considerations about the technical aspects of turbomachinery. In the following I provide some remarks to help the authors in their revision process.

 - Introduction

The introduction is not suitable for this journal: there are only lines 14-22 discussing about turbomachinery, and what is written is not entirely correct from a technical point of view (the kinetic energy does not exert any force!). Please revise and extend discussing other papers in literature discussing turbine wakes measured with optical methods (an example is Zaccaria and Lakshminarayana 1997).

 - Expetimental set-up

Only lines 49-51 are dedicated to the description of the turbine stage. Please provide more info, for example the expansion ratio, the rpm, the flow rate, the peak Mach number at the stator and rotor exit. If available, please discuss the flow at the stator exit, without that it is impossible to provide a convincing interpretation of the flow at the rotor exit, considered the prominent role of the rotor wakes

 - Results

In figure 2, please mark clearly the positions where you think there are the stator wakes 

Regarding expression (2) it is surprising that the authors propose the same quantity (the flow angle) defined in different ways (from tangential or form axial) in different parts of the paper; please make it uniform (claiming a 'limitation' of Atan2 is not acceptable in a scientific paper!).

Authors should consider to eliminate the measurements taken for R = 445; if they do not trust these measurements, it is preferred not to show them: it creates confusion in the reader and do not add anything to the discussion.

Lines 124-127: an anomaly arises on the asymmetry detected for R 460. But the authors do not provide any comment. Please provide an attempt of explanation.

Line 147: how did the authors estimate the turbulent integral length scale?

Figures 10 and 12: the spatial turbulence spectra are, in my opinion, among the best findings of this paper (alongside figure 9); please comment on the deviations that appear, for some cases, from the theoretical trend (-5/3) for the highest wake number.

The off-design analysis must be complemented by turbomachinery-related considerations, at least the variation on rotor-incidence and on the reaction degree. The impact on these features on the rotor wakes must be discussed in combination to the results reported in Figure 11.

 - Discussion

Most readers would expect an analysis on the turbomachinery flow, instead it is an analysis of the measurement issues. It is relevant to this paper, very interesting and fair, please only specify in the title that it is a discussion the measurement issues.

It is, anyway, strange to see these considerations only at the end of the paper. Please create a connection between this section and section 2.3 and specify the set-up used (glass, DT, and statistical quality) in section 2.3.

 - Conclusions

It should contain some turbomachinery - related considerations. The closure on the lack of experts in computational studies should be avoided. The reader is not very interested on organization issues of the teams, please leave such considerations out.

As a final consideration, the paper is well written but please do a proof-reading since the referee has found a number of typos and sentences not correct from the grammar point of view.

Author Response

Response to reviewers

First of all, we want to thank all reviewers for their work with reading our manuscript and for their comments.

Second, we want to apologize to editors and reviewers for late response.

We modified the introduction and we added an interesting observation about degree of anisotropy and the spatial distribution of flatness (fourth statistical moment) of tangential velocity.

Two of three reviewers wish the “discussion” section in front of the “results” section. We exchanged them, and we renamed it as “Issues related to the data quality”.

We added the list of symbols and abbreviations.

We apologize You, but we did not calculate the Integral lengthscale explicitly, because we are not sure, how to do it in a systematic and repeatable way (it strongly depends on the reference point, the correlation is calculated with). We will think about it and maybe we add it up to next round of the review.

We added an artistic combination of data and manual sketch of the system of stator and rotor wakes into the footnote 2. We do not want to disturb any figure containing data, but we feel that it is important to sketch what we are talking about across the entire paper.

On behalf of author collective

Daniel Duda

Reviewer 3 Report

The paper describes experimental investigations of transport of blade wakes through an axial turbine in a central passage away from the endwalls. The PIV investigations are made for a nominal load as well as for a low and high load. The results exhibit velocity and turbulence characteristics in key sections showing also interaction of stator and rotor wakes. The paper is interesting and informative and can be published after some editorial improvements.

  • Paper’s English is generally all right, however a delicate touch is needed to remove singular/plural mistakes, wrong tense / words and other grammatic or stylistic errors.
  • I suggest captions for figures be rewritten to be more concise. Parts of text from the captions can be moved to the main body of the text. I recommend the description of the results is slightly improved.
  • A list of symbols used in the paper is recommended at the beginning of the paper.
  • Conclusions – an expression “…probably due to the smaller area of wake cross-sections, although the wakes themselves are wider…” sounds a bit clumsy although its sense is generally understood.  
  • Conclusions – another expression “…but at this moment we lack an expert for numerical simulations in our team” may be true but it does not belong to the summary of results of investigations. I recommend it is withdrawn from Conclusions.

Author Response

First of all, we want to thank You for Your work with reading our manuscript and for Your comments. We very appreciate Your positive statement. 

Second, we want to apologize to editors and reviewers for late response.

We modified the introduction and we added an interesting observation about degree of anisotropy and the spatial distribution of flatness (fourth statistical moment) of tangential velocity.

Two of three reviewers wish the “discussion” section in front of the “results” section. We exchanged them, and we renamed it as “Issues related to the data quality”.

We added the list of symbols and abbreviations.

We added an artistic combination of data and manual sketch of the system of stator and rotor wakes into the footnote 2. We do not want to disturb any figure containing data, but we feel that it is important to sketch what we are talking about across the entire paper.

On behalf of author collective

Daniel Duda

Round 2

Reviewer 1 Report

Unfortunately, the main point of criticism is still valid - this paper does not show anything really new nor are the results interpreted properly. It can be acknowledged that the authors have certainly spent a lot of work and effort, but still there is nothing that would improve the understanding of turbine flows. Interestingly, there are some deviations from the current state of knowledge, but these are nowhere explained, so it is possible that they can rather be attributed to measurement errors.

With regard to the fact that the glass between the flow passage and the casing has been removed, the results must be considered as doubtful. This can not be resolved by the comparison of results with and without glass as given in Fig. 2 - the radius of the measurement location used is not given, and from Fig. 8 it seems that a measurement at the blade tip, i.e. at the other end of the blading, has been used for this comparison. This does not contribute to the credibility of the work.

Altogether, there are definitely numerous plots prepared with a wide variety of methods, however, there is no proper analysis or discussion of the effects seen.

With regard to the literature cited, the authors should have a good look at the publications from Göttlich and Woisetschläger from the years between 2004 - 2008 and beyond.

Author Response

Dear reviewer,
first of all, I have to apologize for long response time, which has been caused by the COVID infection and quite long quarantine after that and thus a “snowball” of obligations accumulated during the time I was out. 

I thank You for Your work with reading our manuscript, I really missed that very interesting literature You recommended me, I am only surprised, that such a successful group does not continue in their research. I have to admit, that I am an amateur in comparison with the mentioned group; however, I think, that I was able to add some new insight into the problem, mainly around the length-scale of fluctuating structures, which is now explored from two different directions: first, the separation of TKE into channels according to the band-pass-filter (which has been included in the previous versions as well) and, second, in the light of the estimation of spatial integral length-scale calculated from the autocorrelation function in every point.

The discussion about the missing glass and its effect is now more deeply discussed and compared to more variants in a separate conference contribution. Therefore, I think, that this article should focus on the overview of observed quantities and this measuring problem is only shortly mentioned with link to that conference paper, which is open-access, thus the interested reader can find the relevant information without making this article too long and boring for someone, who do not need measurement details.

I completely skipped the angle of average velocity in order to avoid chaos caused by the non-linear arcus-tangent function. On the other hand, the chaos with letter u used for both, axial air velocity component past the rotor and circumferential mid-span rotor velocity, continues.

On behalf of author collective
Daniel Duda

Reviewer 2 Report

The reviewer appreciates the revisions made by the authors in the introduction section, however many remarks about the results were neither considered in the revision nor challenged in the rebuttal.

I still believe that the results section can be improved, so I re-propose the remarks that I made in the first review and that have been ignored.

 - Regarding expression (2) it is surprising that the authors propose the same quantity (the flow angle) defined in different ways (from tangential or form axial) in different parts of the paper; please make it uniform (claiming a 'limitation' of Atan2 is not acceptable in a scientific paper!).

 - Lines 124-127: an anomaly arises on the asymmetry detected for R 460. But the authors do not provide any comment. Please provide an attempt of explanation.

 - The off-design analysis must be complemented by turbomachinery-related considerations, at least the variation on rotor-incidence and on the reaction degree. The impact on these features on the rotor wakes must be discussed in combination to the results reported in Figure 11.

I also add a final remark: when discussing other measurement techniques in the introduction, you cite multi-hole probes and hot wires. Please consider also FRAPP among the class of instrumentation relevant for turbomachinery. 

Author Response

[Figures are in attachment]

Dear Reviewer,
first of all, I have to apologize for long response time, which has been caused by the COVID infection and quite long quarantine after that and thus a “snowball” of obligations accumulated during the time I was out. 
I thank You for Your work with reading our manuscript. I completely remove the angle, as it causes only troubles (it is difficult to average it, there are multiple understandings of zero (axial, tangential with rotor, tangential contra rotor) and it is not clear, how it is affected by the missing radial component), instead I plot directly the velocity components - that’s clear.
The reason for different shape of PDF at plane R460 is a mystery for me. I have several hypotheses about the reason: H1: instrumental noise. I filtered the data more carefully and I did not observe any significant change of the shape, see the figure A1 in attachment.

H2: There is mixing of two regimes due to the large-scale secondary flow (i.e. part of snapshots has different pattern than another part of them). This would lead to a “doublepeak”, however such a doublepeak might be hidden by the broadening of individual snapshot statistics. Therefore, I calculate average of each snapshot and plot such distribution without some evidence of doublepeak (, yellow line in Figure A2, note the statistics is poor, thus there is noise on histogram). What do the rest? That decomposition we call Agrawal decomposition or Spatial Reynolds decomposition and it displays almost same shape as the non-decomposed data do (orange line in Figure A2). Therefore, mixing of two regimes does not play a role. The decreasing of skewness, when applying temporal Reynolds decomposition, suggests, that the source lies within the spatial flow pattern (the time-average is subtracted).

We can ask, where (spatially) lies the source? Let’s look at the spatial distribution of skewness at each point. This statistical moment reveals, if the skewed shape of distribution correlates with some average flow pattern. From Figure A3 we see, that the high positive skewness originates in the stator wakes and that its peak values are stronger in the plane R 460, while surprisingly in the hub plane, the ensemble is not skewed much, while in the upper planes, the positive skewness is more or less balanced by the negative one in rotor jets. This is interesting observation and we added this plot into the manuscript, but it does not response Your question, why it is so. It is caused by the jet oscillations, which bring more deflected velocities into the areas of wakes. But we are not sure, maybe You have a better idea?      

Your question about turbomachinery-related parameters I delegated to Ing. Martin Němec, who is expert in classical turbomachinery.
I added some note about the FRAP method with link to some literature (If You would like cite some specific literature, I can add the citation, no problem).
Sincerely
Daniel Duda
